# A Closer Look at Codistillation for Distributed Training

## Abstract

Codistillation has been proposed as a mechanism to share knowledge among concurrently trained models by encouraging them to represent the same function through an auxiliary loss. This contrasts with the more commonly used fully-synchronous data-parallel stochastic gradient descent methods, where different model replicas average their gradients (or parameters) at every iteration and thus maintain identical parameters. We investigate codistillation in a distributed training setup, complementing previous work which focused on extremely large batch sizes. Surprisingly, we find that even at moderate batch sizes, models trained with codistillation can perform as well as models trained with synchronous data-parallel methods, despite using a much weaker synchronization mechanism. These findings hold across a range of batch sizes and learning rate schedules, as well as different kinds of models and datasets. Obtaining this level of accuracy, however, requires properly accounting for the regularization effect of codistillation, which we highlight through several empirical observations. Overall, this work contributes to a better understanding of codistillation and how to best take advantage of it in a distributed computing environment.

## 1 Introduction

Several recent improvements in the performance of machine learning models can be attributed to scaling the training of neural network models (He et al., 2016; Goyal et al., 2017; Vaswani et al., 2017; Devlin et al., 2018; Shoeybi et al., 2019; Huang et al., 2019; Kaplan et al., 2020; Lepikhin et al., 2020; Brown et al., 2020). Most approaches to scaling up training leverage some form of data parallelism (using multiple workers to compute gradients on different training samples in parallel), and the most common approach to data-parallel training is synchronous first-order optimization.

In synchronous data-parallel training, several replicas of a neural network model are created, each on a different worker. The workers process different mini-batches locally at each step using an optimizer such as Stochastic Gradient Descent (SGD) or Adam (Kingma & Ba, 2015), and the replicas synchronize (i.e., average either gradients or parameters) at every step by communicating either with a centralized parameter server (Li et al., 2014) or using `all_reduce` (Goyal et al., 2017). More computing resources can be used in parallel by increasing the number of workers, effectively increasing the batch size used to compute a stochastic gradient. Increasing the batch size reduces the gradient's variance and ideally makes it possible to increase the learning rate in proportion to the number of workers. By doing so, the number of steps required to reach a given model quality is also reduced in proportion to the number of workers, and a near-linear speedup is achieved (Goyal et al., 2017). However, it has been observed that the linear learning rate scaling strategy leads to performance degradation for very large batch sizes (Goyal et al., 2017), and even with more principled learning rate scaling mechanisms, synchronous SGD with larger batches eventually yields diminishing returns (Johnson et al., 2020).

Synchronous data-parallel methods ensure that all models are precisely synchronized at every step during training. This incurs substantial communication overhead, which increases with the number of replicas, and can quickly become a bottleneck limiting the processing units' utilization (e.g., GPU or TPU), especially when devices communicate over commodity interconnects such as Ethernet. A number of approaches have been proposed to reduce communication overhead, including using mixed-precision (Jia et al., 2018) or other forms of compression (Alistarh et al., 2017; Bernstein

et al., 2018), reducing the frequency of synchronization to not occur after every optimizer step (Stich, 2018; Yu et al., 2019), using gossip-based methods for approximate distributed averaging (Lian et al., 2017; Assran et al., 2019; Wang et al., 2020), or using some combination thereof (Wang & Joshi, 2018; Koloskova et al., 2020).

Codistillation is an elegant alternative approach to distributed training with reduced communication overhead (Anil et al., 2018). Rather than synchronizing models to have the same weights, codistillation seeks to share information by having the models represent the same function (i.e., input-output mapping). Codistillation accomplishes this by incorporating a distillation-like loss that penalizes the predictions made by one model on a batch of training samples for deviating from the predictions made by other models on the same batch. In practice, a worker updating one model can compute the predictions made by another model by reading checkpoints of the other model and performing an additional forward pass. Previous work has demonstrated that codistillation is quite tolerant to asynchronous execution using stale checkpoints, e.g., using another model's checkpoint from up to 50 updates ago without observing a significant drop in accuracy (Anil et al., 2018).

Anil et al. (2018) focuses on the very large batch setting. For example, when training a ResNet-50 on ImageNet, codistilling two models and using batch size 16k per model achieves substantially better performance than training with synchronous SGD and batch size 32k, although the final accuracy is still significantly lower than that achieved by synchronous SGD with a smaller batch size (e.g., 8k or smaller). This performance boost is attributed to an ensembling-like effect introduced by the codistillation loss.

In this paper we study codistillation at moderate batch sizes, i.e., before the performance of synchronous SGD begins to degrade. We demonstrate that it is possible to use codistillation in this regime without losing accuracy. For example, when training a ResNet-50 on ImageNet, we show that codistilling two models, each model using a batch size of 256, achieves comparable performance to training a single model using synchronous SGD with batch size 512. Furthermore, this holds across a range of batch sizes.

Achieving this performance parity involves modifying the way that explicit regularization is used in conjunction with codistillation. This modification stems from new insights into codistillation. Specifically, we demonstrate that codistillation has a regularizing effect. Thus, while increasing the batch size in synchronous SGD helps training by reducing the gradient variance, we conjecture that codistillation helps in a complementary manner via this regularization. Because it has a regularizing effect, care needs to be taken when using codistillation in conjunction with other forms of regularization, such as L2 regularization (weight decay), to avoid over-regularizing. We also evaluate the sensitivity of codistillation to different hyper-parameters like the frequency of reading new checkpoints and learning rate schedule.

Overall, our findings complement previous work on codistillation (Anil et al., 2018; Zhang et al., 2018). We summarize below our main contributions:

1. To the best of our knowledge, we demonstrate for the first time that models trained with codistillation can perform as well as models trained with traditional parallel SGD methods even when trained with the same number of workers and same number of updates, despite using a much weaker synchronization mechanism (Section 3). Previous work at the intersection of codistillation and distributed training used extremely large batch sizes and more workers than the parallel SGD counterparts.

2. Complementing the existing work on codistillation, we show that codistillation acts as a regularizer (Section 4). Our work demonstrates that explicitly accounting for its regularization effect is a key ingredient to using codistillation without losing accuracy (compared to parallel SGD methods).

## 2 CODISTILLATION: BACKGROUND AND RELATED WORK

Codistillation is proposed as a mechanism for sharing information between multiple models being trained concurrently (Anil et al., 2018; Zhang et al., 2018). In typical multi-phase distillation, first, a teacher model is trained using standard supervised learning, and then a student model is trained to predict the outputs of the teacher model without any updating of the teacher. In contrast, when two

---

**Algorithm 1:** Codistillation

---

**Input:** Loss function $L(y, \hat{y})$ and codistillation loss function $D(y, y')$
**Input:** Model architecture $f_\theta(x)$ and initial model parameters $\{\theta_i^1 : i = 1, \ldots, n\}$
**Input:** Number of iterations $K$, learning rates $\{\eta^k\}_{k=1}^K$, and penalty coefficients $\{\alpha^k\}_{k=1}^K$

1 **for** $k = 1, \ldots, K$ **do**
2     **for** $i = 1, \ldots, n$ **do**
3        $x, y = \texttt{get\_next\_minibatch()}$
4        $\theta_i^{k+1} = \theta_i^k - \eta^k \nabla_{\theta_i} \left( L(y, f_{\theta_i^k}(x)) + \alpha^k \frac{1}{n-1} \sum_{j \neq i} D\left( f_{\theta_i^k}(x), f_{\theta_j^k}(x) \right) \right)$
5     **end**
6 **end**

---

or more models *codistill*, there is only one phase, and in addition to minimizing the usual supervised loss on the training data, an additional loss term is used to share information between models by encouraging each model to make similar predictions to the other(s).

Codistillation, as described in Zhang et al. (2018), is shown in Algorithm 1. Here, $n \geq 2$ models are trained concurrently. The $i$th model is updated on line 4 by taking a gradient step to minimize the combination of a standard supervised loss function $L$ (e.g., cross-entropy or MSE) and a distillation-like loss $D$ which penalizes differences between the predictions made by model $i$ and those made by model $j$, averaged over all other models $j \neq i$. Zhang et al. (2018) and Anil et al. (2018) both report using Kullback-Liebler (KL) divergence for $D$ in their experiments, although Anil et al. (2018) mentions that other options are possible (e.g., mean squared error between the logits of different models). Zhang et al. (2018) does not explicitly include a penalty parameter $\alpha^k$, and instead (implicitly) takes $\alpha^k = 1$ for all $k$.

Anil et al. (2018) focuses on codistillation as a mechanism for scaling distributed training, and demonstrates that using codistillation allows for better scaling with larger batch sizes than conventional synchronous parallel SGD. When different models reside on different GPUs, implementing line 4 requires that the devices communicate. Anil et al. (2018) proposes to have workers exchange parameter checkpoints, and these checkpoints are updated periodically rather than after every iteration. This leads to some delay, i.e., actually using $\theta_j^{k'}$ with $k' < k$ in line 4. However, it is argued that such delay is tolerable for codistillation because large changes in parameters $\theta$ do not necessarily correspond to large changes to the function $f_\theta$. Experimentally, it is reported that exchanging checkpoints every 50 updates does not noticeably degrade performance in their setup. Anil et al. (2018) also reports using a burn-in period since, "in the beginning of training, the distillation term in the loss is not very useful or may even be counterproductive." During the burn-in period of the first $\hat{K} \ll K$ steps, the models train independently; i.e., $\alpha^k = 0$ for $k = 1, \ldots, \hat{K}$. Below we find that codistillation can tolerate large delays (e.g., updating checkpoints after hundreds or thousands of updates) without any burn-in period and without any significant loss in accuracy.

One common application of distillation is to train a single model to mimic an ensemble's predictions to avoid the computational overhead of deploying an ensemble. Motivated by this, previous works (Anil et al., 2018; Zhang et al., 2018) explore the connection between codistillation and ensemble methods, showing that two models trained using codistillation achieve performance close to that of ensembling two independently-trained models. Anil et al. (2018) also conducts an experiment that compares codistilling two models with simply using label smoothing as a regularizer when training one model, and reports that codistillation performs significantly better.

## 3    Codistillation, synchronous SGD, and regularization

The aim of this section is to demonstrate that codistillation can be used as a mechanism for distributed training without losing accuracy, even when used in conjunction with batch sizes that are not particularly large. To achieve this, we also show that regularization needs to be handled carefully. Typical training regimes use some form of regularization to avoid overfitting. For example, the common practice is to use L2 regularization (weight decay) when training convolutional networks (Goyal et al., 2017) and to use label smoothing when training neural machine translation

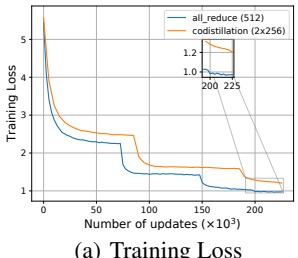 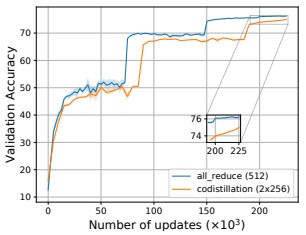 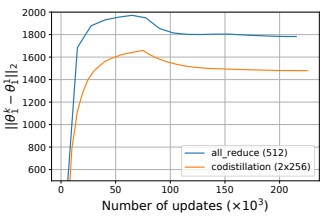

(a) Training Loss  (b) Top-1 Validation Accuracy  (c) Change in parameter values from initialization

Figure 1: Comparing `all_reduce` and codistillation for training ResNet50 on ImageNet. We codistill two models using batch size 256 for each, and the model trained with `all_reduce` uses batch size 512. All experiments are run on 16 GPUs. We report the (a) training loss and (b) top-1 validation accuracy. We observe that compared to `all_reduce`, the model trained with codistillation underfits, obtaining higher training loss and lower top-1 accuracy. (c) Examining the difference in parameters from initialization over the course of training further suggests that codistillation has a regularizing effect that impacts performance, since parameters remain closer to their initial values.

models (Ott et al., 2018). Directly incorporating codistillation on top of existing training pipelines can lead to underfitting. We begin this section by illustrating this point.

## 3.1 CODISTILLATION "OUT OF THE BOX" CAN OVER-REGULARIZE

We train a ResNet50 (He et al., 2016) model to perform image classification on the ImageNet dataset (Russakovsky et al., 2015). To begin, we compare training a single synchronous SGD model using batch size 512 with codistilling two models using batch size 256 per model. In both cases we perform training using 16 GPUs and batch size 32 per GPU. Similar to Anil et al. (2018), we follow the training procedure proposed in Goyal et al. (2017) for the learning rate warmup, learning rate schedule, L2 regularization, and other hyperparameters. Specifically, synchronous SGD training (`all_reduce`) is run for 90 epochs. The learning rate starts at 0.1, regardless of batch size, and is warmed up over the first five epochs to $0.1 \times b/256$, where $b$ is the effective batch size across all workers (e.g., 512 when using synchronous SGD and 16 GPUs). The learning rate is then decreased by a factor of 0.1 at the beginning of epochs 30, 60, and 80.

When training with codistillation, workers read the checkpoints of the other model once every 2500 updates. At this frequency of checkpoint reading, we did not observe any benefit to using a burn-in period. We experimented with reading checkpoints more frequently and also did not observe any improvement in performance. When training with two models and batch size 256, we use the same total number of GPUs (16) as an `all_reduce` model with batch size 512, as well as the same number of gradient updates per GPU (with each update processing a mini-batch of size 32). Hence, when training using codistillation we only train the two models for 45 epochs (so that the number of updates per codistilled model is the same as the number of updates performed by `all_reduce` using the same total number of GPUs) and scale all milestones in the learning rate schedule by half, i.e.., warming up over the first 2.5 epochs, and decreasing at epochs 15, 30, and 40.

In Fig. 1(a) and 1(b), we observe that the model trained with codistillation achieves higher training loss and lower top-1 accuracy (around 2% lower) compared to the model trained with synchronous SGD. The codistillation performance is calculated using one of the two models (model 1) being codistilled. This suggests that although the codistillation loss may provide a useful training signal, without other modifications it is not sufficient to achieve the same accuracy as synchronous SGD using the same number of GPUs.

Since both training loss and validation accuracy are affected, we hypothesize that the problem stems from over-regularization. To further investigate this we plot the change in model weights, relative to their initialization, over the course of training in Fig. 1(c). We observe that parameters do not move as far in parameter space when training with codistillation as they do when using synchronous SGD, providing further evidence that codistillation acts as a regularizer.

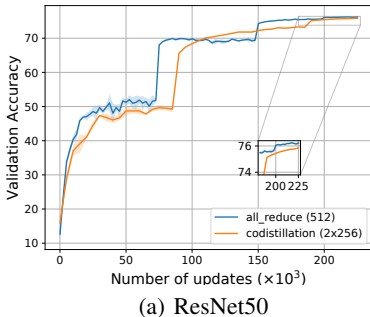 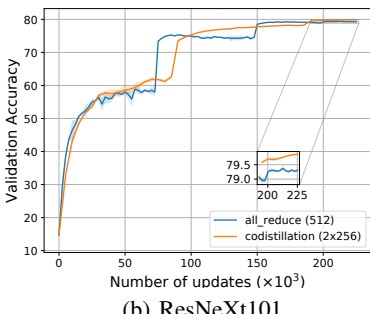

(a) ResNet50            (b) ResNeXt101

Figure 2: Comparing `all_reduce` and codistillation for ResNet50 and ResNeXt-101 models on the ImageNet dataset, with decreasing weight decay and a shifted learning rate decay schedule compared to Goyal et al. (2017). Both methods achieve similar values of top-1 validation accuracy.

## 3.2    ACCOUNTING FOR CODISTILLATION'S REGULARIZATION EFFECT BRIDGES THE GAP

Motivated by the observations above, we conduct an experiment using less explicit regularization when training with codistillation. Goyal et al. (2017) recommends using a constant L2 regularization of $10^{-4}$ throughout training. We propose to start with the same initial value ($10^{-4}$), reduce it to $10^{-5}$ after the first learning rate decay, and reduce it further to $0$ after the second learning rate decay. When training with codistillation, we observe that the model's training loss saturates slower and we shift the learning rate schedule by a few epochs to account for this (from 15, 30, 40 to 18, 38, 44).

In Fig. 2(a), we compare the top-1 validation accuracy on ImageNet for the ResNet50 model trained with codistillation and `all_reduce`, with the above-mentioned modifications to the codistillation setup. Accounting for codistillation's regularization effect helps narrow the performance gap, with `all_reduce` now performing only marginally better ($76.1\%$ for `all_reduce` vs $75.9\%$ for codistillation). We verify that with a constant L2 regularization of $10^{-4}$ throughout training, codistillation performs much worse than `all_reduce`, implying that the improved performance is not only from the modified learning rate schedule; see Fig. 12 in the Appendix. To validate that the improvement is not specific to the ResNet50 architecture, we repeat the experiment using the larger ResNeXt101, ResNet152 (Fig. 9 in Appendix) and ResNext152 (Fig. 10 in Appendix) architectures. In Fig. 2(b), we observe that codistillation performs marginally better than `all_reduce`. We believe that the larger ResNeXt101 may benefit more from codistillation due to having more capacity than the ResNet50. The training loss curves for both models are shown in Fig. 8 in the Appendix.

In synchronous SGD (`all_reduce`), more workers can be added to increase the effective batch size (summed across all workers). This reduces the gradient's variance and the model can be trained with a larger learning rate and fewer steps, while maintaining a similar level of accuracy (Goyal et al., 2017). In Fig. 3 and Table A.2, we demonstrate a similar effect with codistillation. As one doubles the batch size per worker (and hence doubles the effective batch size), the learning rate can also be doubled and the model reaches similar performance in half the number of steps.

The experiments above use the step-wise learning rate schedule described in Goyal et al. (2017). We want to ascertain that our findings are not dependent on this specific learning rate schedule. Hence we train the ResNet50 and ResNeXt101 models with the cosine learning rate schedule (He et al., 2019). In Fig. 4, we observe that the final validation performance for the two approaches is very close, confirming that codistillation works consistently across different learning rate schedules. The corresponding training loss plots are shown in Fig. 11 in the Appendix.

So far, all our experiments have focused on the ImageNet dataset and the ResNet family of models. Next, we evaluate the codistillation mechanism for neural machine translation (NMT). Specifically, we train the "big" transformer model (Vaswani et al., 2017) (6 blocks in the encoder and decoder networks) on the WMT'16 En-De translation dataset, following the setup described in Ott et al. (2018). We note that for codistillation, reducing the explicit regularization is again important for achieving performance comparable to `all_reduce` (see details in Appendix, Section A.1).

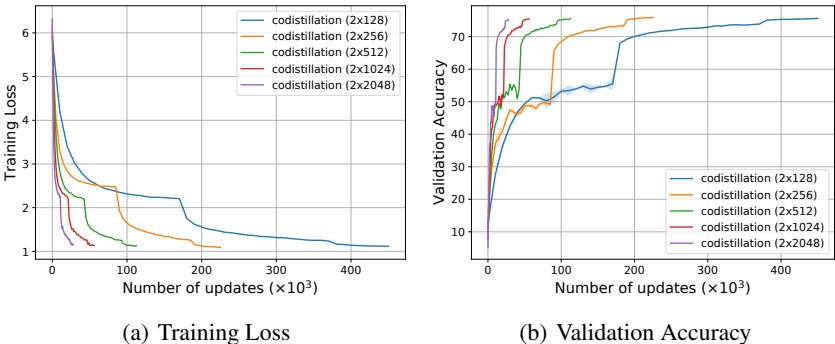

(a) Training Loss  (b) Validation Accuracy

Figure 3: Codistillation scales well across multiple values of batch size per worker. Each time we double the batch size per worker, we scale the learning rate schedule by a factor of two and perform half the number of updates. We do not observe any significant degradation in (a) training loss or (b) validation accuracy across a wide range of batch sizes. The training loss and validation accuracy for different batch sizes are also provided in Table A.2.

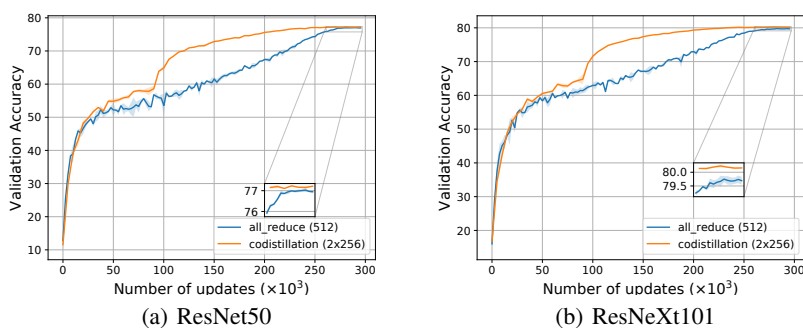

(a) ResNet50  (b) ResNeXt101

Figure 4: Comparing `all_reduce` and codistillation for ResNet50 and ResNeXt101 models on the ImageNet dataset, using a cosine learning rate schedule (He et al., 2019). We observe that the final validation performance for the two approaches is very close, confirming that codistillation works consistently across different learning rate schedules.

This observation is in line with our previous observations on convolutional models. In Fig. 5, we observe that the model trained with codistillation reaches a similar validation loss as the model trained with `all_reduce`. This confirms that codistillation with adjusted regularization schedule also extends to NMT. The corresponding training plot is shown in Fig. 14 in the Appendix.

There are several important aspects of the above results that are worth emphasizing. **(i)** Unlike prior work on codistillation, we train both setups with the same number of workers for the same number of steps. We believe this constraint to be particularly important to better understand the impact of codistillation on many practical distributed training workflows. **(ii)** Our objective is not to show that codistillation can outperform `all_reduce`, and our initial

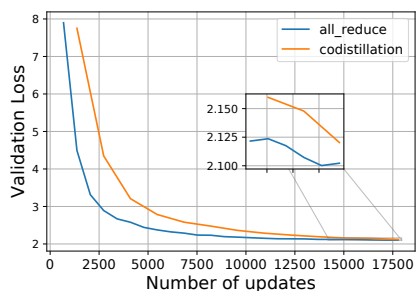

Figure 5: Comparing `all_reduce` and codistillation using "big" transformer model on WMT'16 En-De dataset. Models trained using `all_reduce` and codistillation both reach similar NLL on the validation dataset.

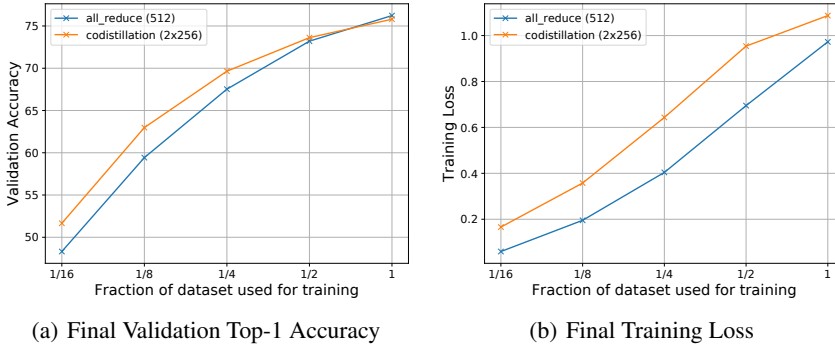

(a) Final Validation Top-1 Accuracy        (b) Final Training Loss

Figure 6: Final validation top-1 accuracy and training loss of a ResNet-50 model when trained using a fraction of the full training data. We observe as smaller fractions of training data are used (and model starts overfitting), codistillation setup increasingly improves over the `all_reduce` setup in terms of validation accuracy.

expectation was actually that replacing `all_reduce` by codistillation would degrade the performance when using the same number of updates (as suggested in Fig. 1), since codistillation is a much weaker form of synchronization. With these experiments, we want to quantify this performance loss and understand how we can minimize it. Interestingly, we find that the performance gap can be narrowed down significantly, with codistillation even outperforming `all_reduce` in some settings (Fig. 4(b)). **(iii)** We believe that previous works did not investigate the regularization effect of codistillation because they focused on the ensembling effect obtained by increasing the number of workers. Being aware of the regularization effect and explicitly accounting for it is important when using codistillation in practice, as seen in Fig. 2.

## 4    CODISTILLATION HELPS REDUCE OVERFITTING

Section 3 illustrated that applying codistillation in addition to other forms of regularization can result in over-regularizing, and progressively reducing the explicit regularization helps improve training and generalization. This section further explores the regularizing effect of codistillation by examining settings where we expect the model to overfit.

We simulate a scenario where the ResNet50 model is likely to overfit in a controlled way by training the model using only a fraction of the full ImageNet training set. When training on $(1/k)$th of the training set, we multiply the number of epochs by $k$ so that the total number of model updates performed is the same as when training with the full training set. The learning rate and weight decay schedules are also modified accordingly. As we train on less data, we expect the model to overfit, i.e., to obtain lower training loss and lower validation accuracy.

In Figure 6, we indeed observe that overfitting occurs when using less training data. However, the overfitting is less severe when training using codistillation, providing further support for the hypothesis that codistillation acts as a regularizer.

An interesting side-effect of this observation is that codistillation could be an interesting alternative to `all_reduce` for training over-parameterized models. We further verify this hypothesis by training a "big" Transformer model for a small NMT dataset (IWSLT 2014 German-English translation dataset that contains 153K training sentences, 7K development sentences, and 7K test sentences). The model trained with codistillation achieves a validation NLL of 2.31 whereas the model trained with `all_reduce` reaches a validation NLL of 2.37.

## 5    ROBUSTNESS TO UPDATE FREQUENCY AND LEARNING RATE

To further understand the codistillation setup, we conduct additional ablations examining the effect of some hyper-parameter choices.

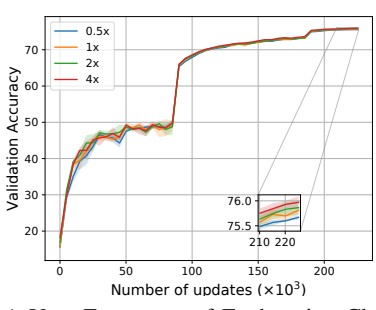 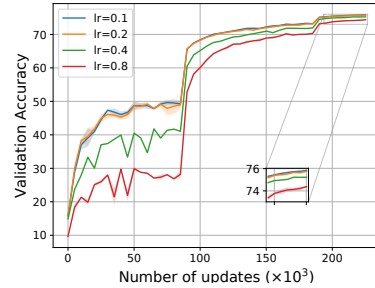

(a) Vary Frequency of Exchanging Check-points

(b) Vary Learning Rate

Figure 7: Evaluating the robustness of codistillation setup by varying the frequency of exchanging checkpoints (left) and learning rate (right) when training ResNet50 on ImageNet dataset. While there are some gains by using a higher frequency, the results do not degrade too much with a lower frequency. The learning rate can be increased from 0.1 to 0.2 without hurting the performance.

In distributed training, model weights are exchanged regularly between the training workers. In synchronous SGD, model parameters are communicated every gradient update, via `all_reduce`, which introduces the large communication overhead. In codistillation, the frequency of reading model checkpoints should be treated as a hyper-parameter and should be tuned for the specific use cases. Zhang et al. (2018) always uses the most recent version of the model for computing the codistillation loss, equivalent to reading a new checkpoint at every iteration. Anil et al. (2018) reports exchanging checkpoints every 50 updates.

In the ImageNet results presented in Sections 3 and 4, we exchange the model checkpoints once every 2500 updates. In Fig. 7(a), we evaluate how the validation performance for the ResNet models varies with the frequency of exchanging the checkpoints). The curve corresponding to $1\times$ is the baseline setup, with checkpoints exchanged every 2500 updates. Label $2\times$ denotes the case where checkpoints are exchanged twice as frequently (every 1250 updates), and other curves similarly show $0.5\times$ and $4\times$. The corresponding training loss curves are shown in Fig. 15(a) in the Appendix. We observe that the setup is quite robust to the frequency of exchange. Although there are some small improvements by using a higher frequency, the results do not degrade too much with a lower frequency. Similar trends hold for the Transformer model (for language translation task), as shown in Fig. 16(a) in the Appendix.

In Fig. 7(b), we vary the base learning rate when training the codistillation model on the ImageNet dataset, using the setup proposed in Goyal et al. (2017) and report the accuracy on the validation dataset. We observe that increasing the learning rate from 0.1 to 0.2 has almost no impact on the validation accuracy. Beyond that, higher learning rates hurt the validation accuracy more. The corresponding plot with the training loss is shown in Fig. 15(b) in the Appendix.

## 6 CONCLUSION

In this work, we demonstrate for the first time (to the best of our knowledge), that even for moderate batch sizes, models trained with codistillation can perform as well as models trained with traditional parallel SGD methods (while using comparable computational resources). Complementing existing works, we show that codistillation acts as a regularizer, and that accounting for this property is essential when using codistillation. There are several exciting and potentially impactful directions for extending our understanding of codistillation. One important direction is to codistill more than two models. The loosely synchronized nature of codistillation potentially allows using different topologies for determining which models codistill together. Zhang et al. (2018) does consider one version of the setup, where all models codistill with each other, but they do so in a non-distributed setting. Another possible future work will be to consider codistillation between models that are different, e.g., in terms of architecture, capacity, or hyper-parameters, and evaluate if that improves performance over the standard knowledge distillation based approaches.

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

## A APPENDIX

### A.1 IMPLEMENTATION DETAILS

The code for all experiments is implemented using PyTorch 1.5 (Paszke et al., 2017). All experiments are run for 3 seeds and plots report the average over all seeds. The codistillation loss is the mean squared error between the logits of the two models (without centering them, as we found in preliminary experiments that centering yielded similar results).

The experiments involving the ImageNet dataset are based on the setup proposed in Goyal et al. (2017). We consider two schedules to control the value of the codistillation loss coefficient $\alpha^k$ – the *constant* schedule, $\alpha^k = 1$ for all $k$, and the *step-wise* schedule where we increase $\alpha^k$ every time we decrease the learning rate. We did not find any significant difference in the performance of the two cases and report the performance for the case where $\alpha^k$ increases by 0.5 every time we decrease the learning rate.

The experiments for translation tasks use the FairSeq library (Ott et al., 2019). These experiments are based on the setup proposed in Ott et al. (2018). We consider two schedules to control the value of $\alpha^k$ - the *constant* schedule and the *exponential* schedule where we increase $\alpha^k$ by a factor $\gamma = 1.1$ every epoch. We find in Fig. 17 that the *exponential* schedule works better than the *constant alpha* schedule (2.12 vs 2.15 validation perplexity), and this is the schedule we use in all other experiments on this task. When training the translation models with codistillation, we explicitly reduce the amount of regularization (to account for regularization added by codistillation) by removing the label smoothing loss.

### A.2 ADDITIONAL RESULTS ON IMAGENET

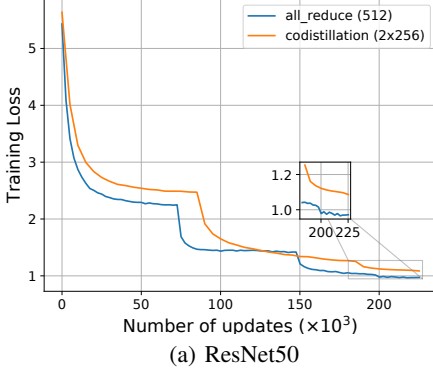
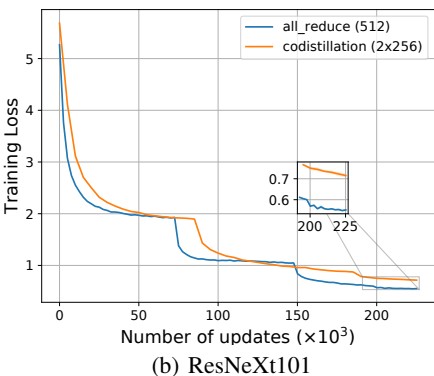

(a) ResNet50

(b) ResNeXt101

Figure 8: Comparing the training loss for the `all_reduce` and codistillation setups for ResNet50 and ResNeXt-101 models (respectively) on the ImageNet dataset, with decreasing weight decay and a shifted learning rate decay scheduled compared to Goyal et al. (2017). While the `all_reduce` setup reaches a lower training loss, the performance on the validation dataset (in terms of top-1 validation accuracy) is very similar for the two setups (for both models) as shown in Fig. 2.

In Fig. 8, we plot the training loss for the ResNet50 (left) and ResNeXt101 (right) models for the ImageNet dataset with some variations to the setup proposed in Goyal et al. (2017), as in Section 3.2. Specifically, Goyal et al. (2017) recommends using a constant $L2$ weight decay set to $10^{-4}$ throughout training. Keeping the initial value of this weight decay to $10^{-4}$, we reduce it to $10^{-5}$ after the first learning rate decay and further to $0$ after the second learning rate decay. We also modify the learning rate schedule from Goyal et al. (2017), which is based on how the training loss changes (and saturates) during training. Due to the regularization effect of codistillation, we observe that the model's training loss saturates slower and we shift the schedule by a few epochs to account for this (from 15, 30, 40 to 18, 38, 44). We note that for both models, the `all_reduce` setup reaches

a lower training loss. Despite this, the performance on the validation dataset (in terms of top-1 validation accuracy) is very similar for the two setups (for both models) as shown in Fig. 2.

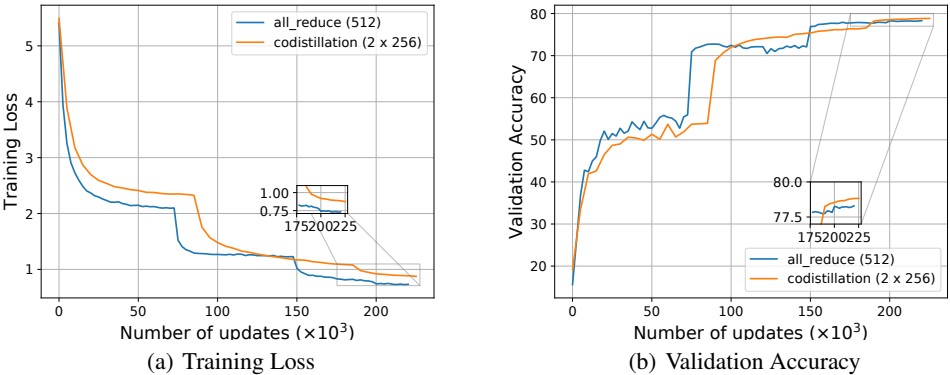

(a) Training Loss

(b) Validation Accuracy

Figure 9: Comparing the training loss and validation accuracy (respectively) for the `all_reduce` and codistillation setups on the ImageNet dataset, with decreasing weight decay and a shifted learning rate decay scheduled compared to Goyal et al. (2017). While the `all_reduce` setup reaches a lower training loss, the performance on the validation dataset (in terms of top-1 validation accuracy) is very similar for the two setups.

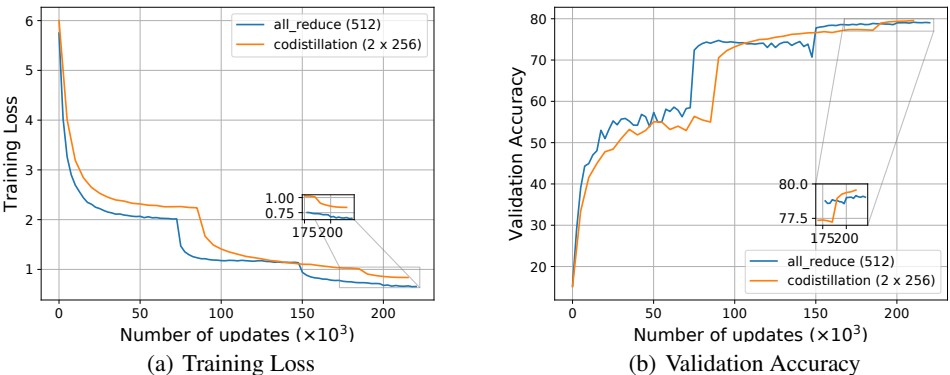

(a) Training Loss

(b) Validation Accuracy

Figure 10: Comparing the training loss and validation accuracy (respectively) for the `all_reduce` and codistillation setups on the ImageNet dataset, with decreasing weight decay and a shifted learning rate decay scheduled compared to Goyal et al. (2017). While the `all_reduce` setup reaches a lower training loss, the performance on the validation dataset (in terms of top-1 validation accuracy) is very similar for the two setups.

In Fig. 11, we show the training performance for ResNet50 and ResNeXt101 models when using the cosine learning rate schedule (He et al., 2019). Again, while the `all_reduce` setup reaches a lower training loss, the performance on the validation dataset (in terms of top-1 validation accuracy) is very similar for the two setups (for both models) as shown in Fig. 4.

In Fig. 12 we verify that the good performance observed in Fig. 2 is not due only to the shifted learning rate schedule. We observe that, indeed, with this shifted schedule, if we keep the $L2$ weight decay to a constant value of $10^{-4}$ throughout training, then codistillation is significantly worse than `all_reduce` both in terms of training loss and validation accuracy.

In synchronous SGD (`all_reduce`), more workers can be added to increase the effective batch size (summed across all workers). The increased batch size reduces the gradient's variance and the

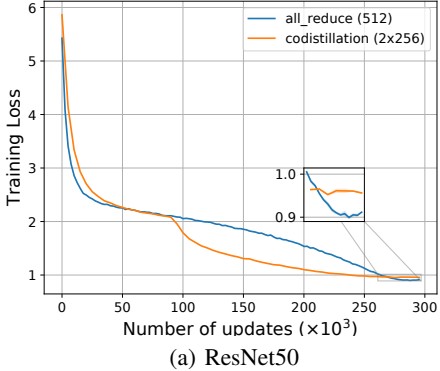 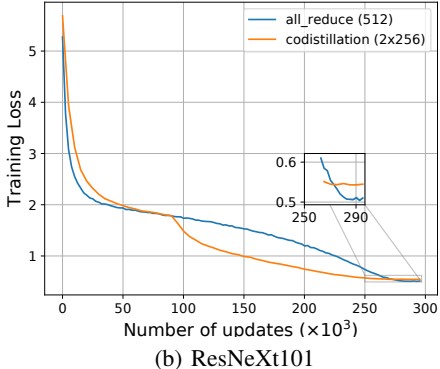

(a) ResNet50            (b) ResNeXt101

Figure 11: Comparing the training loss for the `all_reduce` and codistillation setups for ResNet50 and ResNeXt-101 models (respectively) on the ImageNet dataset, using the cosine learning rate schedule proposed in He et al. (2019). While the `all_reduce` setup reaches a lower training loss, the performance on the validation dataset (in terms of top-1 validation accuracy) is very similar for the two setups (for both models) as shown in Fig. 4.

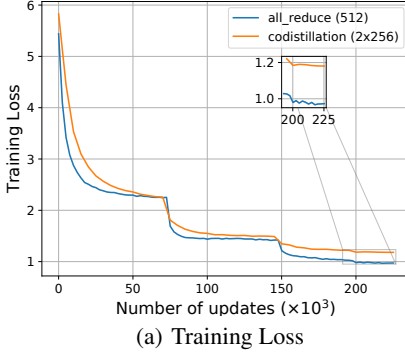 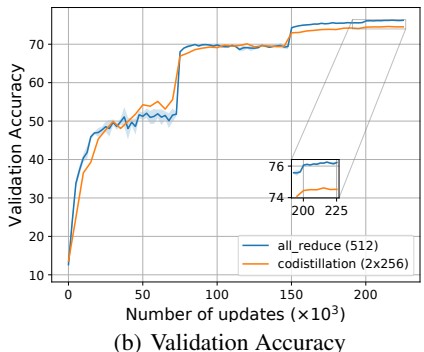

(a) Training Loss            (b) Validation Accuracy

Figure 12: Comparing the training loss (left) and validation accuracy (right) of the `all_reduce` and codistillation setups with the ResNet50 model on the ImageNet dataset. The training setup is the same as in Fig. 2(a), with one difference – the $L2$ weight decay is kept at a constant value of $10^{-4}$ throughout training. These plots show that when we do not account for the regularization effect of codistillation, the codistillation model's performance lags behind `all_reduce`.

model can be trained with a larger learning rate and fewer steps, while maintaining a similar level of accuracy (Goyal et al., 2017). In Fig. 3 and Table A.2, we demonstrate a similar effect with codistillation. As one doubles the batch size per worker (and hence doubles the effective batch size), the learning rate can also be doubled and the model reaches similar performance in half the number of steps.

| Batch Size | Training Loss | Validation Accuracy |
|---|---|---|
| $2 \times 128$ | 1.12 | 75.61 |
| $2 \times 256$ | 1.08 | 75.92 |
| $2 \times 512$ | 1.12 | 75.51 |
| $2 \times 1024$ | 1.13 | 75.45 |
| $2 \times 2048$ | 1.14 | 75.26 |

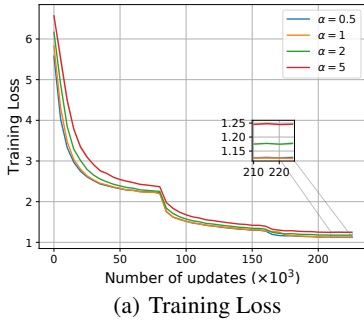 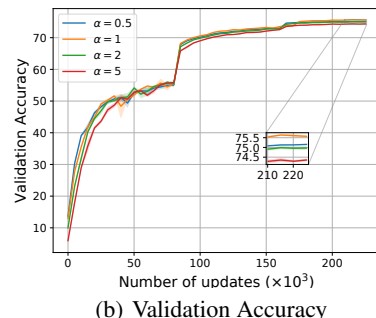

(a) Training Loss         (b) Validation Accuracy

Figure 13: Evaluating codistillation setup by varying the value of $\alpha_k$. Note that the value is not changed during training and hence $\alpha_k$ is denoted by $\alpha$.

## A.3 Effect of varying $\alpha$

In Fig 13, we evaluate codistillation setup by varying the value of $\alpha_k$. Note that the value is not changed during training and hence $\alpha_k$ is denoted by $\alpha$.

## A.4 Additional results on WMT'16 En-De

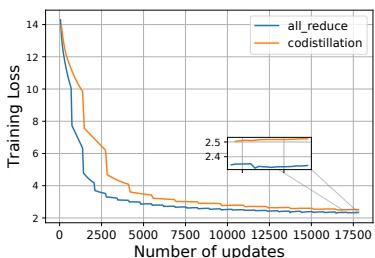

Figure 14: Comparing `all_reduce` and codistillation setups using the "big" transformer model on the WMT'16 En-De dataset. The model trained using codistillation performs worse in terms of training performance, but generalizes well to the validation dataset, as seen in Fig. 5.

In Fig. 14 we plot the training loss of the `all_reduce` and codistillation setups on the WMT'16 En-De dataset. We observe that codistillation has a higher training loss, but almost matches the validation loss of `all_reduce` as shown in Fig. 5.

## A.5 Additional results on codistillation's robustness

To further understand the codistillation setup, we conduct additional ablations examining the effect of some hyper-parameter choices.

In Fig. 15(a), we compare the training performance of ResNet50 models trained on the ImageNet dataset, as we vary the frequency of exchanging the checkpoints. The curve corresponding to $1\times$ is the baseline setup, with checkpoints exchanged every 2500 updates. Label $2\times$ denotes the case where checkpoints are exchanged twice as frequently (every 1250 updates), and other curves similarly show $0.5\times$ and $4\times$. The training loss is robust to changes to the frequency, reaching very similar values across the whole range. The corresponding top-1 validation accuracy is shown in Fig. 7(a).

In Fig. 15(b), we vary the base learning rate when training the codistillation model on the ImageNet dataset, using the setup proposed in Goyal et al. (2017) and report the loss on the training dataset.

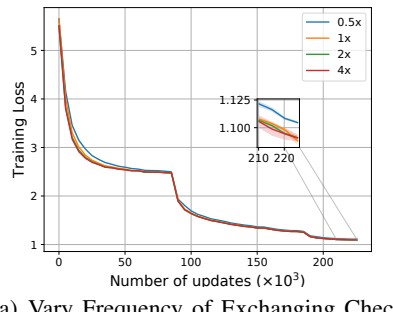 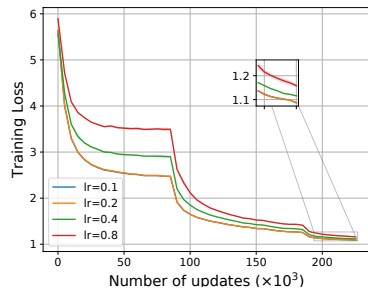

(a) Vary Frequency of Exchanging Checkpoints

(b) Vary Learning Rate

Figure 15: Evaluating the training loss robustness of the codistillation setup by varying the frequency of exchanging checkpoints (left) and learning rate (right) when training a ResNet50 model on the ImageNet dataset. The training loss is robust to changes to the frequency, reaching very similar values across the whole range. The learning rate can be increased from 0.1 to 0.2 without hurting the training performance.

We observe that increasing the learning rate from $0.1$ to $0.2$ has almost no impact on the training loss. Beyond that, higher learning rates affect the training loss more. The corresponding plot with the top-1 validation accuracy is shown in Fig. 7(b) and exhibits a similar trend.

In Fig. 16, we vary the frequency of exchanging the model checkpoints for the Transformer model. We note that more frequent updates lead to a higher validation loss initially, but as training continues, this setup eventually outperforms (slightly) the one with less frequent updates. Overall, the model's performance does not decline drastically even when we exchange checkpoints 2.5 times less often.

In Fig. 17, both the initial value of $\alpha$ (codistillation loss coefficient) and $\gamma$ (scaling factor for $\alpha$) are varied when training the Transformer model. Since we are using a geometric progression for increasing the value of $\alpha$, changing $\gamma$ alone can increase $\alpha$ to very large values, thus making training unstable. Hence, for each $\gamma \in \{1, 1.1, 1.5\}$ we vary the initial $\alpha$ to achieve optimal results. The case of $\gamma = 1$ corresponds to keeping $\alpha$ constant. This case slightly under-performs as compared to the cases where we vary $\alpha$.

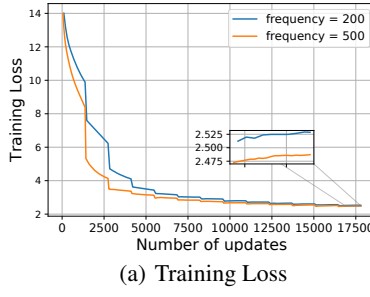 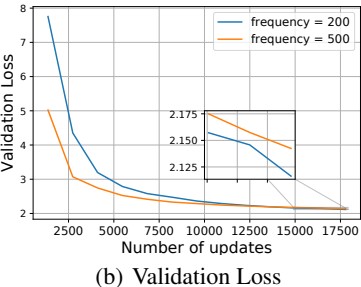

(a) Training Loss

(b) Validation Loss

Figure 16: Evaluating the robustness of codistillation for the Transformer model by varying the frequency of exchanging checkpoints. The performance does not degrade when exchanging checkpoints less frequently.

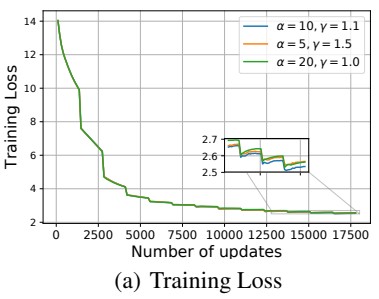

(a) Training Loss

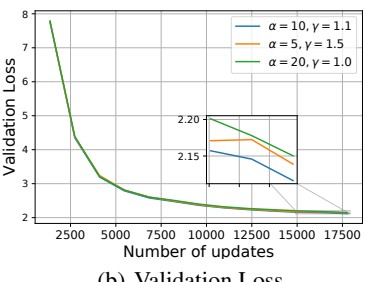

(b) Validation Loss

Figure 17: Evaluating the robustness of codistillation for the Transformer model by varying the initial value of $\alpha$ and $\gamma$. Given the nature of geometric progression, we perform a sweep over $\alpha$ to achieve optimal performance for each $\gamma \in \{1, 1.1, 1.5\}$. The case of $\gamma = 1$ corresponds to keeping $\alpha$ constant. This case slightly under-performs as compared to the cases where we increase $\alpha$ over time.

