# OpenReview forum: "A Closer Look at Codistillation for Distributed Training"
_ICLR.cc/2021/Conference — Reject_

### Official Review · AnonReviewer2 · 2020-10-21
**The paper investigates codistillation as an alternative to distributed synchronous gradient descent, by comparing on batch sizes and learning rates that are typical for the latter. A hypothesis is put forward that codistillation acts as a regularizer, and evidence is provided showing that gradually decreasing the model's built-in regularizers (e.g. L2) over training epochs allows codistillation to more closely match the achieved accuracy of synchronous distributed training.**

**Rating:** 4
**Confidence:** 4

**Review:**

Update: I have read the authors' response, and have decided to keep my score as-is

Claims:

1) Codistillation achieves accuracy close to synchronous gradient descent if the model's built-in regularizers (e.g. L2) are decreased to zero while training

2) Codistillation therefore acts as a regularizer, and should be explicitly accounted for during training

3) Codistillation works for moderate batch sizes, not just large-batch

Pros:

-Interesting observations about codistillation's empirical behavior, which contributes to the overall literature on distributed training

-Codistillation imposes much lower network costs since model parameters only have to be exchanged once every few thousand updates (rather than every mini-batch of updates)

Cons:

-There are problems with the systems setting, which undermine the strength and validity of the claims:

1) The setting is not fully clear: given that only n=2 codistillation replicas are used throughout the paper, how are the GPUs arranged? My guess is that there are 2 physical machines with 8 GPUs each, and within each machine, the GPUs exchange their updates synchronously. This information should be clearly presented.

2) The overheads of synchronous distributed training increase with higher machine/replica count, so codistillation should be really be studied with many replicas, which is where it has the greatest potential to have an advantage. Note that, depending on the model, network and GPU hardware, synchronous distributed training on just 2 replicas/machines may not even incur any network overhead, which completely eliminates the overhead-reducing advantages of codistillation. Hence, the paper needs to study codistillation with at least n=4 to 16 replicas, verifying that the regularization effect of codistillation still holds with more replicas, and that the proposed means of controlling regularization effects (namely, reducing the built-in regularizers) is still effective. Thought experiment: what happens if the regularization effect of having 16 replicas is so strong (because there are 16 terms in the auxiliary loss) that reducing the built-in regularizers is no longer sufficient?

3) The increased compute overhead of codistillation (due to auxiliary loss needing to evaluate every of the n model replicas every update) is not investigated, so it is unclear how it stacks up against the decrease in network costs. Runtime benchmarks versus all-reduce could have been provided and analyzed.

-Experiments are performed on only two models, ResNet and Transformer. Comparable papers that focus on empirical evidence usually test their hypotheses on five or more models, in order to demonstrate that the technique is truly generic. Testing on non-residual CNN architectures (e.g. VGG) and RNN architectures would have made the results more convincing.

-No systematic method is proposed/evaluated for reducing the model's built-in regularizers. Even if no theory is presented, I imagine the paper could have presented at least a rate function with controllable hyperparameter (similar to learning rate), and systematically conducted experiments for different values of the hyperparameter. Currently, each model's builtin regularizers are reduced in an ad hoc fashion, so the evidence is only of anecdotal quality.

Suggestions for improvement:

-In Section 4, the authors may want to clarify whether the model's built-in regularizers (L2 etc.) are or are not being reduced - it was not clearly stated, and I am guessing it is the latter case.

-The paper would not be complete without a discussion of Federated Learning (McMahan et al., 2017), which provides another means of reducing the network overhead of synchronous distributed training, among other benefits. Even if experiments against Federated Learning are not conducted, it should be discussed in the related work.

---

> ### Author Response · Authors · 2020-11-13
> **Thank you for your constructive feedback**
>
> We thank the reviewer for their constructive feedback.  We address their concerns below:
>
> 1. Regarding "The setting is not fully clear: ... This information should be clearly presented.", we will update the description in the paper to improve clarity. For all the experiments, we use n=2 codistillation replicas. The replicas are not restricted to using just 8 gpus. For example, In Figure 5, each replica uses 64 gpus. In Figure 3, each curve corresponds to different number of gpus per replica
>
> 2. Regarding "The overheads of synchronous distributed training ... means of controlling regularization effects (namely, reducing the built-in regularizers) is still effective. ", we agree with the point that whether or not codistillation may be advantageous depends on the hardware/system (including interconnect) as well as the task (model architecture, dataset). Previous work (Anil et al.[1]) provided evidence that codistillation can be faster than minibatch SGD. Our focus in this paper is on demonstrating that codistillation can achieve accuracy on par with a well-tuned, well-established mini-batch SGD baseline, which was not considered in previous work. As mentioned above, we do considere experiments using up to 64 GPUs, and increasing the number of replicas is not the only way to leverage additional GPUs. We also agree that investigating the use of more replicas is an important direction, and we have left this as future work.
>
> 3. Regarding "Thought experiment: what happens if the regularization effect of having 16 replicas is so strong (because there are 16 terms in the auxiliary loss) that reducing the built-in regularizers is no longer sufficient", we thank the reviewer for suggesting this interesting thought experiment. There are several ways to keep the regularization effect in check. We could compute the mean of the codistillation loss (across replicas) or we can make $\alpha$ smaller.
>
> 4. Regarding "The increased compute overhead ... provided and analyzed.", we agree  with the reviewer that it will be interesting to compare models in terms of wallclock time. Our current implementation is not optimized for speed/wallclock time. In this work, we focus on understanding codistillation and evaluating if codistillation can be used as an alternative for all_reduce, without losing accuracy. The answer to this question is not obvious given that codistillation uses a much weaker form of synchronization.
>
> 5. Reagrding "Comparable papers ... more convincing", we thank the reviewer for their suggestion and are working on adding VGG results for ImageNet.
>
> 6. Regarding "No systematic method ... anecdotal quality", we thank the reviewer for suggesting this interesting experiment. We are working on adding a plot with different values of L2 regularization.
>
> 7. Regarding "In Section 4, ... is the latter case.", we thank the reviewer for bringing this up. The regularizers are reduced from 10^-4 to 10^-5 to 0 whenever the learning rate is decayed.  We will update this section in the paper.
>
> 7. Regarding "the paper would not be complete without a discussion of Federated Learning ... (discussed in the related work.", we thank the reviewer for their suggestion and will update the paper with references to Federated Learning.
>
>
> Thank you again for the comments and suggestions. Please let us know if we have addressed your concerns.
>
>
> References
>
>
> [1]:  Rohan Anil, Gabriel Pereyra, Alexandre Passos, Robert Ormandi, George E. Dahl, and Geoffrey E. Hinton. Large scale distributed neural network training through online distillation. In International Conference on Learning Representations (ICLR), 2018.
>
> [2]: Ying Zhang, Tao Xiang, Timothy M. Hospedales, and Huchuan Lu. Deep mutual learning. In IEEE Conf. on Computer Vision and Pattern Recognition (CVPR), pp. 4320–4328, 2018.

---

> ### Author Response · Authors · 2020-11-17
> **Update on new experiments**
>
> We are thankful for your constructive feedback and comments.
>
> We have run some experiments, with the VGG models. We are using the VGG16 model implementation from [0]. The all_reduce model achieves about 70.1 accuracy while the model trained with codistillation archives around 69.7 % accuracy.
>
> We have run some more experiments, with larger vision models (ResNet151 and ResNext 151). In both the cases, we observe that models trained with codistillation perform as well as the models trained without codistillation:
>
> For ResNet151, codistillation trained models obtain 78.63 vs all_reduce model obtains about 78.35 (close to the value of 78.31 reported at [1]). For ResNext151, codistillation trained models obtain 79.57 vs all_reduce model obtains about 79.13 (close to the value of 79.1 reported at [2]).
>
> We will be updating the paper and putting the corresponding plots in the appendix.
>
> The first phase of response period is going to end soon. If you could indicate any other concerns you have, we will be happy to address them.
>
> [1]: https://pytorch.org/docs/stable/torchvision/models.html
> [2]: https://github.com/facebookresearch/pycls/blob/master/MODEL_ZOO.md

---

> ### Author Response · Authors · 2020-11-21
> **Updated the paper**
>
> Dear Reviewer
>
> We thank you for your constructive feedback and have updated the paper with the following additional results (in the Appendix):
>
> 1. In Fig 9 and 10 (page 12), we have added results for two new models: ResNet152 and ResNext 152. In both the cases, we observe that models trained with codistillation perform as well as the models trained without codistillation.  For ResNet152, codistillation trained models obtain 78.63 vs all_reduce model obtains about 78.35 (close to the value of 78.31 reported at [1](https://pytorch.org/docs/stable/torchvision/models.html)). For ResNext151, codistillation trained models obtain 79.57 vs all_reduce model obtains about 79.13 (close to the value of 79.1 reported at [2](https://pytorch.org/docs/stable/torchvision/models.html)).
>
> 2. In Fig 13 (page 14), we have added an ablation result for different values of $\alpha$.
>
> 3. In Table A2, (page 13), we have tabulated the results from Fig 3 (page 6) for making the comparison easier. We note that codistillation scales well across multiple values of batch size per worker.
>
> We thank you for your comments and suggestions. Please let us know if we have addressed your concerns.

---

### Official Review · AnonReviewer4 · 2020-10-28
**Incremental empirical work on co-distillation for distributed training with weak evaluation**

**Rating:** 4
**Confidence:** 4

**Review:**

This work analyzes the effect of co-distillation for distributed training under moderate batch sizes. Using distillation-like techniques to improve synchronous SGD training is an interesting direction. And the paper carefully analyzed this setting while using the same amount of compute, which is not done by prior work to my knowledge. In addition, the writing is good and easy to follow.

However, I think the work can be improved in the following aspects:

W1: The setup of the work is not very realistic.

The authors proposed to analyze distributed training (using two machines) with very small batch sizes (e.g., 256). These two constraints don't appear to be commonly used for ML tasks where distributed training help. E.g., for CV, Exploring the Limits of Weakly Supervised Pretraining (Dhruv et al., 2018) was already using batch sizes of 8,064 back in 2018; modern NLP tasks use significantly larger batch sizes (up to 4M in GPipe, Huang et al., 2018). Even if hardware is a limiting factor, showing how results scale with more than 2 model copies would help a lot in demonstrating the potential value of this work in distributed settings.

W2: The contribution of this work seems to be incremental.

Considering distillation / co-distillation's regularization effect is not exactly a novel direction. This line of thoughts originated at least from Distilling the Knowledge in a Neural Network (Hilton et al., 2014) and was discussed in other papers like Collaborative Learning for Deep Neural Networks (Song et al., 2018).

W3: Experiments could be done on top of stronger baselines.

The authors proposed that co-distillation benefits models with higher capacity more (e.g., 3.2: ResNeXt10(1) may benefit more from codistillation due to having more capacity than ResNet50), but there are no CV experiments demonstrating that this scales with more parameters done on more complex architectures (e.g., SENet, AmoebaNet, etc).

W4: In the NLP/NMT case, there are no experiments showing that this benefits self-attention architectures besides a remark in Section 4. For small datasets like IWSLT'14 DE->EN, it's known that using fewer parameters  (e.g., 256d) vs the default 1024 in transformer-big produces much better results. See e.g., Tied Transformers: Neural Machine Translation with Shared Encoder and Decoder (AAAI'19). Comparing the regularization effect with 6 layers / 1024 dimension as a baseline setup feels like cherry picking.  For NMT it's also much more common to report BLEU vs perplexity.

W5: if the paper's goal is to demonstrate co-distillation as a viable distributed training alternative, it would be also valuable to compare it with other related training methods like gossip SGD etc.


====== Edit after author response ====

I have read authors' responses and have decided to keep my score as is.

The additional experiments haven't addressed problem formulation issues (W1, W2, W5) if the paper is positioned as more of a theoretical work; if the paper is positioned as a more of an experimental work, the baselines used (W3, W4) need to be improved with proper hparams settings.

---

> ### Author Response · Authors · 2020-11-13
> **Thank you for the constructive feedback.**
>
> We thank the reviewer for their constructive feedback and for the comment "the paper carefully analyzed this setting while using the same amount of compute, which is not done by prior work to my knowledge.". We address their concerns below:
>
> 1. Regarding w1 "The authors proposed to analyze distributed training (using two machines) with very small batch sizes (e.g., 256)", we highlight the results in Figure 3, where we consider using different batch sizes (256, 512, 1024, 2048). These experiments use up to 8 machines with 8 GPUs each. Regarding use of very large batch sizes, our work focuses on the moderate batch regime. Previous work (Anil et al [1]) have shown benefits of codistillation in a very large batch regime (e.g., 16K on ImageNet). Our motivation to study moderate batch sizes is to verify that it achieves performance comparable to a well-tuned minibatch SGD baseline, while the previous work focused on highlighting the potential of codistillation on tasks which either have not been well-studied in the literature (so there was not a strong baseline) or used batch sizes so large that performance is known to be degraded.
>
> 2. Regarding W2 "Considering distillation / co-distillation's regularization effect is not exactly a novel direction.", we respectfully disagree with the reviewer due to following reasons:
>
> * In distillation, the student model learns from a trained teacher model, while in codistillation, both models can be seen as both teachers and students.
> * In Song et al the distillation loss is applied only on the classifier head.
> * Previous works in codistillation which are more closely related to our work ([1, 2]) do not report observing any regularization effect, thus making our findings useful and novel.
>
> 3. Regarding W4 and W5, our focus is on understanding codistillation and showing that codistillation can be used as a mechanism for distributed training without losing accuracy. We believe that this finding, along with our analysis of the regularization effect that explains it, is interesting and novel enough by itself. In this paper we do not claim that codistillation is the best distributed training algorithm that exists, though we do hope that our work will motivate further research along this direction..
>
>
> Thank you again for the comments and suggestions. Please let us know if we have addressed your concerns.
>
>
> References:
>
> [1]:  Rohan Anil, Gabriel Pereyra, Alexandre Passos, Robert Ormandi, George E. Dahl, and Geoffrey E. Hinton. Large scale distributed neural network training through online distillation. In International Conference on Learning Representations (ICLR), 2018.
>
> [2]: Ying Zhang, Tao Xiang, Timothy M. Hospedales, and Huchuan Lu. Deep mutual learning. In IEEE Conf. on Computer Vision and Pattern Recognition (CVPR), pp. 4320–4328, 2018.

---

> ### Author Response · Authors · 2020-11-17
> **Update on new experiments**
>
> We are thankful for your constructive feedback and comments.
>
> We have run some additional experiments, with larger vision models (ResNet151 and ResNext 151). In both the cases, we observe that models trained with codistillation perform as well as the models trained without codistillation:
>
> For ResNet151, codistillation trained models obtain 78.63 vs all_reduce model obtains about 78.35 (close to the value of 78.31 reported at [0]).  For ResNext151, codistillation trained models obtain 79.57 vs all_reduce model obtains about 79.13 (close to the value of 79.1 reported at [1]).
>
> We will be updating the paper and putting the corresponding plots in the appendix.
>
> The first phase of response period is going to end soon. If you could indicate any other concerns you have, we will be happy to address them.
>
> [0]:https://pytorch.org/docs/stable/torchvision/models.html
>
> [1]: https://github.com/facebookresearch/pycls/blob/master/MODEL_ZOO.md

---

> ### Author Response · Authors · 2020-11-21
> **Updated the paper**
>
> Dear Reviewer
>
> We thank you for your constructive feedback and have updated the paper with the following additional results (in the Appendix):
>
> 1. In Fig 9 and 10 (page 12), we have added results for two new models: ResNet152 and ResNext 152. In both the cases, we observe that models trained with codistillation perform as well as the models trained without codistillation.  For ResNet152, codistillation trained models obtain 78.63 vs all_reduce model obtains about 78.35 (close to the value of 78.31 reported at [1](https://pytorch.org/docs/stable/torchvision/models.html)). For ResNext151, codistillation trained models obtain 79.57 vs all_reduce model obtains about 79.13 (close to the value of 79.1 reported at [2](https://pytorch.org/docs/stable/torchvision/models.html)).
>
> 2. In Fig 13 (page 14), we have added an ablation result for different values of $\alpha$.
>
> 3. In Table A2, (page 13), we have tabulated the results from Fig 3 (page 6) for making the comparison easier. We note that codistillation scales well across multiple values of batch size per worker.
>
> We thank you for your comments and suggestions. Please let us know if we have addressed your concerns.

---

### Official Review · AnonReviewer1 · 2020-10-29
**Non-surprising explanation about the role of codistillation in distributed training with empirical observations**

**Rating:** 4
**Confidence:** 4

**Review:**

This paper aims to have a closer look at the role of codistillation for distributed training. Authors provided an answer with their empirical observations. That is, codistillation acts as a regularizer, since the distance between the learned model and the initialization is smaller than sync SGD without codistillation. Then, the authors claim that the codistillation may over-regularize and study how to modify the training configurations to avoid it. There are further discussions on the overfitting and robustness to hyper-parameters in sec 4 and sec 5.


Comparing to sync SGD, codistillation bring in auxillary loss enforcing all the local workers learn from each other in some degree. Thus, it is intuitively a regularization term. Deep understanding on how it takes effect will help us better trade off the bias and variance in distributed training. Unfortunately, this paper does not achieve this goal.

Most of the claims in this paper are not surprising, with limited insights. 1) It is common sense that, regularization may “over-regularize” and can “reduce overfitting”. The empirical observations to demonstrate these are not necessary in sec 3.1 and sec 4. 2) To “bridge the (over-regularize) gap” (In sec 3.2), authors report the empirical behavior of distributed training with codistillation, with difference choice of learning rate, batch size, concise learning rate schedule, tasks (NMT), and update frequency (in sec 5). The proposed modification of training is minor for a specific setting, i.e., to decreasing learning rate from 10^{-4} to 10^{-5} on ImageNet for ResNet50.

Authors ignore the factors which are more direct to trade off bias and variance. 1) The coefficient of the auxiliary loss \alpha can adjust the regularization. I have not seen the discussion on it. What is the value for it in the experiments? Did I miss anything? 2) For me, the number of local workers may alleviate or enlarge the regularization of codistillation. It seems that, most of the experiments are conducted with 2 models/workers.

Overall, this paper is easy to follow, but the results (from the experiments) are non-surprising, with limited insights. The modification proposed is minor and did’t get significant improvement of performance.


Overall, this paper proposed methods to make CNN to be symmetric as (NS euqation) dynamics. The technical contribution need be highlighted.

---

> ### Author Response · Authors · 2020-11-13
> **Thank you for the constructive feedback (Part 1/2)**
>
> We thank the reviewer for their constructive feedback and address their concerns below:
>
>
> 1. "Comparing to sync SGD, codistillation bring in auxillary loss enforcing all the local workers learn from each other in some degree. Thus, it is intuitively a regularization term”
>
> Although we agree that it intuitively looks like a regularization term, we believe that its regularization effect is actually far from obvious. We can see at least the three following reasons why codistillation is not obviously a regularizer. Each model is being distilled against one (or several) other models that vary over time, while in the standard distillation setup they do not vary (and typical regularization schemes like L2 regularization penalize a distance to a fixed point as well). It is particularly important to note that when other models have the same architecture and optimize the same supervised training loss (as in our setup), then the global optimum of the optimization task augmented with codistillation is actually the same as without it! (Indeed, when all models are at this same optimum, the supervised loss is minimized and the codistillation loss is zero). This contrasts with traditional regularization schemes that change the global optimum of the optimization task. There could be situations where adding the codistillation loss actually leads to more overfitting. As an example, consider codistilling two models where model #0 does not use any L2 regularization, while model #1 does. Model #0 may tend to overfit due to lack of regularization, which could also cause model #1 to overfit (more than it would have without codistillation). We will add a discussion to this effect in the paper as it is indeed an important topic to address.
>
> 2. We agree with the reviewer that “It is common sense that, regularization may “over-regularize” and can “reduce overfitting””. We respectfully disagree with the reviewer’s assumption that codistillation “obviously” has a regularizing effect, as discussed above. The empirical observations to demonstrate these are indeed necessary in sec 3.1 and sec 4.
>
> 3. Regarding "To “bridge the (over-regularize) gap” learning rate schedule, tasks (NMT), and update frequency (in sec 5)", We did not understand the reviewer's comment. Are they saying that reporting “ the empirical behavior of distributed training with codistillation, with different choice of learning rate, batch size, concise learning rate schedule, tasks (NMT), and update frequency (in sec 5)” is not required? Our objective was to show that codistillation can be used as an alternative for synchronous SGD for different tasks, setups etc.
>
> 4. Regarding “The proposed modification of training is minor for a specific setting,  i.e., to decreasing learning rate from 10^{-4} to 10^{-5} on ImageNet for ResNet50“. We agree with the reviewer that the proposed modification is a minor change, but:
> * We believe the reviewer meant to say “weight decay” instead of “learning rate”
> * The more general observation is that one may need to reduce the effect of other regularizers when using codistillation, and this observation holds beyond ImageNet as we found it useful as well on the NMT task. See Appendix A.1: “When training the translation models with codistillation, we explicitly reduce the amount of regularization (to account for regularization added by codistillation) by removing the label smoothing loss”.
>  * We disagree with the reviewer in terms of the impact of this minor change. Specifically, our focus is on showing that codistillation can be used as a mechanism for distributed training without losing accuracy. We show that this is indeed achievable, with only small changes in the hyperparameters that are tuned to work well with the all_reduce setup. We also provide an explanation to justify these changes: the regularization effect of codistillation, which we highlight through several empirical observations.
>
>
> 5. Regarding “The coefficient of the auxiliary loss \alpha can adjust the regularization. I have not seen the discussion on it.”, we will update the paper with a plot with different values of \alpha for the ResNet models. A corresponding curve for Transformer models is provided in Figure 14 in Appendix. Specifically, we observe that for different values of $\alpha$, the model can reach similar levels of performance.

---

> ### Author Response · Authors · 2020-11-13
> **Thank you for the constructive feedback (Part 2/2)**
>
> 6. Regarding “What is the value for it in the experiments? Did I miss anything?”, we provide the value of this hyper-parameter in section A1 (in Appendix).  Copy-pasting from the appendix “We consider two schedules to control the value of the codistillation loss coefficient α^k -- the constant schedule, α^k = 1 for all k, and the step-wise schedule where we increase α^k every time we decrease the learning rate. We did not find any significant difference in the performance of the two cases.“
>
> 7. Regarding “the modification proposed is minor and didn't get significant improvement of performance”. We reiterate that the objective of our work is not to show that codistillation can outperform all_reduce, and our initial expectation was actually that replacing all_reduce by codistillation would degrade the performance when using the same number of updates (as suggested in Fig. 1), since codistillation is a much weaker form of synchronization. The fact that codistillation-trained models can perform as well as models trained with all_reduce, with “minor” modifications, is a strength of our work.
>
> Minor remark: the last sentence of the review looks like a copy/paste leftover that should probably be removed.
>
> Thank you again for the comments and suggestions. Please let us know if we have addressed your concerns.
>
> References:
>
> [1]: Rohan Anil, Gabriel Pereyra, Alexandre Passos, Robert Ormandi, George E. Dahl, and Geoffrey E. Hinton. Large scale distributed neural network training through online distillation. In International Conference on Learning Representations (ICLR), 2018.
>
> [2]: Ying Zhang, Tao Xiang, Timothy M. Hospedales, and Huchuan Lu. Deep mutual learning. In IEEE Conf. on Computer Vision and Pattern Recognition (CVPR), pp. 4320–4328, 2018.a

---

> ### Author Response · Authors · 2020-11-17
> **Anything else you would like us to respond to?**
>
> We are thankful for your constructive feedback and comments.
>
> The first phase of response period is going to end soon. If you could indicate any other concerns you have, we will be happy to address them.

---

> ### Author Response · Authors · 2020-11-21
> **Updated the paper**
>
> Dear Reviewer
>
> We thank you for your constructive feedback and have updated the paper with the following additional results (in the Appendix):
>
> 1. In Fig 9 and 10 (page 12), we have added results for two new models: ResNet152 and ResNext 152. In both the cases, we observe that models trained with codistillation perform as well as the models trained without codistillation.  For ResNet152, codistillation trained models obtain 78.63 vs all_reduce model obtains about 78.35 (close to the value of 78.31 reported at [1](https://pytorch.org/docs/stable/torchvision/models.html)). For ResNext151, codistillation trained models obtain 79.57 vs all_reduce model obtains about 79.13 (close to the value of 79.1 reported at [2](https://pytorch.org/docs/stable/torchvision/models.html)).
>
> 2. In Fig 13 (page 14), we have added an ablation result for different values of $\alpha$.
>
> 3. In Table A2, (page 13), we have tabulated the results from Fig 3 (page 6) for making the comparison easier. We note that codistillation scales well across multiple values of batch size per worker.
>
> We thank you for your comments and suggestions. Please let us know if we have addressed your concerns.

---

### Official Review · AnonReviewer3 · 2020-11-06
**Interesting Results on Codistillation**

**Rating:** 5
**Confidence:** 2

**Review:**

Summary of paper:
The paper studies the concept of codistillation in data parallel distributed training. In this setting, the standard minibatch SGD algorithm requires exchange of models in every update of every node. Recent work in distributed training has studied "local SGD", where models are exchanged at frequent (usually periodic) intervals after a bunch of local updates. This paper studies an alternative, called "codistillation". The idea is that at a given node, say node $i$, the local model updates are regularized by the most recent models at nodes $\{j, j \neq i\}$ through an appropriately modified loss term. Specifically, the loss term bias the model at node $i$ towards having similar classification outcomes on the training data as the (most recent) local estimate of the model at nodes $\{j, j \neq i\}.

The paper argues that this codistillation approach has  a regularizing effect, and if the other regularization approaches are combined with codistillation in a balanced manner so as to avoid over-regularization, then the performance can be similar to minibatch SGD, with much lower global communication.

Pros:
--> The paper's results are certainly intriguing and likely to lead to further investigations of codistillation as an alternative or complementary approach to local SGD for reducing communications in distributed training.
 --> The results are promising and the experimental results seem quite comprehensive.

Cons:
--> I am not sure completely understand/buy the reasoning provided in the paper. The regularization in imagenet - as I understand it - biases towards chosing models with lower weights (e.g., sparser models). The regularization effect of co-distillation is towards chosing models that are not too far away from the initial point - indeed 1(c) shows that the codistilled models are not too far away from the original models.  However, the paper (e.g, Sections 3.1, 3.2) seems to suggest these two forms of regularization are similar/equivalent, and one form can be increased and the other form can be lowered.  I find this explanation not very convincing.
--> I think the regularization parameters (e.g., what exactly is the constant L2 regularization, what is the factor of 10^{-4} in Section 3.2) needs to be explained in more detail in the paper for completeness...at this point, the reading requires knowledge of the details of Goyal (2017) for the reader to know what exactly is implemented by the authors.
--> The result that codistillation with local updates can lead to lower loss as compared to minibatch SGD is almost too surprising to be true. Common intuition suggests that the best achievable performance in terms of accuracy per update that any distributed approach can get is that of minibatch SGD, which is what one would do if all the data was centralized at one node. So I am suspcious of the improvements. I think there are two possible reasons, that the authors might clarify (or suggest other reasons)
a) The benchmark minibatch SGD model is not regularized properly, or something else is ineffectively done in this benchmark estimate (e.g., perhaps hyper parameters are not as well chosen).
b) Perhaps I do not understand what exactly is done in the minibatch SGD setting, e.g., is the minibatch SGD done with one round of global communication in every model update, or is there some adjustment made to make the communication costs comparable?

Summary:
Although not all aspects of the paper are convincing,  the results of the paper are intriguing and it is likely to lead to further investigations along this topic.  I am on the borderline on this paper, perhaps leaning slightly more towards acceptance.


====== Comments after author response ====

Thanks for the detailed response. Having read the update and the other reviews, I have lowered my score.
 I am still left unconvinced on (at least) two aspects.

--> It is unclear to me as to how a method can improve upon minibatch-SGD can actually have better generalization. While nothing theoretically rules this out, it possible means that that either (A) the model architecture was sub-optimal and there is room for improvement, or (B) the optimization found a minimum that is not very good. In either case, there appears to be a different "lesson"  than the one described in the paper. This is certainly worth exploring in more detail.

--> I agree with some of the other reviewers that the experimental results can be improved, specifically, the usage of just 2 replicas (now clarified in the paper) is limiting, and a more detailed analysis regarding how to tune the regularization parameters.

I do think that the paper is on track towards an interesting discovery, but I would like to see a deeper/more detailed analysis to be convinced.

---

> ### Author Response · Authors · 2020-11-13
> **Thank you for the constructive feedback**
>
> We thank the reviewer for their constructive feedback and address their concerns below:
>
> 1. "the paper (e.g, Sections 3.1, 3.2) seems to suggest these two forms of regularization are similar/equivalent, and one form can be increased and the other form can be lowered."
>
> We clarify that we are not suggesting that the two regularizations are equivalent. Our argument is that using codistillation (along with the L2 regularizer) leads to over regularization of the model and we try to address that by reducing the effect of L2 regularization.
>
> 2. "At this point, the reading requires knowledge of the details of Goyal (2017) for the reader to know what exactly is implemented by the authors."
>
> We apologize for the lack of details and will update the paper. To address the specific question here, Goyal et al. uses a regularization loss of the form regularization_constant * L2 loss. The regularization_constant is set to the standard default value of 10^{-4} as suggested in Goyal et al.
>
> 3.  "best achievable performance in terms of accuracy per update that any distributed approach can get is that of minibatch SGD"
>
> We agree with this intuition when considering other distributed approaches that aim to approximate minibatch SGD (e.g., gossip methods, local SGD, quantizing or compressing communications) but with reduced communication. However, codistillation is not doing this, and we don’t see any reason to expect that minibatch SGD is fundamentally superior in terms of accuracy per update.
>
> 4. "The benchmark minibatch SGD model is not regularized properly, or something else is ineffectively done in this benchmark estimate (e.g., perhaps hyper parameters are not as well chosen)"
>
> We respectfully disagree. For the mini batch SGD baseline, we use the same setup as proposed in Goyal et al[1]. As a sanity check, we note that we report 79.38% top-1 accuracy with the ResNext101 model while the pretrained model in PyTorch model zoo reports 79.31% accuracy. Similarly, for ResNet50, we report top-1 accuracy of 76.1% which matches the pretrained model (76.15%). (https://pytorch.org/docs/stable/torchvision/models.html)
>
> We also emphasize that the objective of our work is not to show that codistillation can outperform all_reduce, and our initial expectation was actually that replacing all_reduce by codistillation would degrade the performance when using the same number of updates (as suggested in Fig. 1), since codistillation is a much weaker form of synchronization. However, we find that by properly accounting for the regularization aspect (which is not reported by the previous papers [2][3]) we can use codistillation without losing accuracy.
>
> 5. "Perhaps I do not understand what exactly is done in the minibatch SGD setting"
>
> We reiterate that for the mini batch SGD baseline, we use the same setup as proposed in Goyal et al.[1], i.e., one round of global communication (all_reduce) per model update.
>
> Thank you again for the comments and suggestions. Please let us know if we have addressed your concerns.
>
> References:
>
> [1]: Priya Goyal, Piotr Dollar, Ross Girshick, Pieter Noordhuis, Lukasz Wesolowski, Aapo Kyrola, An- ´
> drew Tulloch, Yangqing Jia, and Kaiming He. Accurate, large minibatch sgd: Training imagenet
> in 1 hour. arXiv preprint arXiv:1706.02677, 2017.
>
> [2]: Rohan Anil, Gabriel Pereyra, Alexandre Passos, Robert Ormandi, George E. Dahl, and Geoffrey E.
> Hinton. Large scale distributed neural network training through online distillation. In International Conference on Learning Representations (ICLR), 2018.
>
> [3]: Ying Zhang, Tao Xiang, Timothy M. Hospedales, and Huchuan Lu. Deep mutual learning. In IEEE
> Conf. on Computer Vision and Pattern Recognition (CVPR), pp. 4320–4328, 2018.

---

> ### Author Response · Authors · 2020-11-17
> **Anything else you would like us to respond to?**
>
> We are thankful for your constructive feedback and comments.
>
> The first phase of response period is going to end soon. If you could indicate any other concerns you have, we will be happy to address them.

---

> ### Author Response · Authors · 2020-11-21
> **Updated the paper**
>
> Dear Reviewer
>
> We thank you for your constructive feedback and have updated the paper with the following additional results (in the Appendix):
>
> 1. In Fig 9 and 10 (page 12), we have added results for two new models: ResNet152 and ResNext 152. In both the cases, we observe that models trained with codistillation perform as well as the models trained without codistillation.  For ResNet152, codistillation trained models obtain 78.63 vs all_reduce model obtains about 78.35 (close to the value of 78.31 reported at [1](https://pytorch.org/docs/stable/torchvision/models.html)). For ResNext151, codistillation trained models obtain 79.57 vs all_reduce model obtains about 79.13 (close to the value of 79.1 reported at [2](https://pytorch.org/docs/stable/torchvision/models.html)).
>
> 2. In Fig 13 (page 14), we have added an ablation result for different values of $\alpha$.
>
> 3. In Table A2, (page 13), we have tabulated the results from Fig 3 (page 6) for making the comparison easier. We note that codistillation scales well across multiple values of batch size per worker.
>
> We thank you for your comments and suggestions. Please let us know if we have addressed your concerns.

---

### Decision · Program_Chairs · 2021-01-07
**Final Decision**

**Decision:**

Reject

**Comment:**

This paper proposes a novel and interesting approach called co-distillation for distributed training. The main idea is to add a regularizer in order to encourage local models to be consistent with the global objective. Although the idea is a promising alternative to local-update SGD, the approach is mostly empirical. The claim that co-distillation helps reduce overfitting could be better justified by theoretical analysis, in addition to the experimental results. I hope that the reviewers' constructive comments will help improve the paper for future re-submission.